# Signatures of hybridization in *Trypanosoma brucei*

**Christopher Kay[1], Lori Peacock[1,2], Tom A. Williams[1], Wendy Gibson[1]***

**1** School of Biological Sciences, University of Bristol, Bristol, United Kingdom, **2** Bristol Veterinary School, University of Bristol, Bristol, United Kingdom

* w.gibson@bristol.ac.uk

**Data Availability Statement:** All data generated or analysed during this study are included in this published article, its supplementary information files and associated online resources listed in S1_Text. The data are available from the Dryad

## Abstract

Genetic exchange among disease-causing micro-organisms can generate progeny that combine different pathogenic traits. Though sexual reproduction has been described in trypanosomes, its impact on the epidemiology of Human African Trypanosomiasis (HAT) remains controversial. However, human infective and non-human infective strains of *Trypanosoma brucei* circulate in the same transmission cycles in HAT endemic areas in subsaharan Africa, providing the opportunity for mating during the developmental cycle in the tsetse fly vector. Here we investigated inheritance among progeny from a laboratory cross of *T. brucei* and then applied these insights to genomic analysis of field-collected isolates to identify signatures of past genetic exchange. Genomes of two parental and four hybrid progeny clones with a range of DNA contents were assembled and analysed by k-mer and single nucleotide polymorphism (SNP) frequencies to determine heterozygosity and chromosomal inheritance. Variant surface glycoprotein (VSG) genes and kinetoplast (mitochondrial) DNA maxi- and minicircles were extracted from each genome to examine how each of these components was inherited in the hybrid progeny. The same bioinformatic approaches were applied to an additional 37 genomes representing the diversity of *T. brucei* in subsaharan Africa and *T. evansi*. SNP analysis provided evidence of crossover events affecting all 11 pairs of megabase chromosomes and demonstrated that polyploid hybrids were formed post-meiotically and not by fusion of the parental diploid cells. *VSG*s and kinetoplast DNA minicircles were inherited biparentally, with approximately equal numbers from each parent, whereas maxicircles were inherited uniparentally. Extrapolation of these findings to field isolates allowed us to distinguish clonal descent from hybridization by comparing maxicircle genotype to *VSG* and minicircle repertoires. Discordance between maxicircle genotype and *VSG* and minicircle repertoires indicated inter-lineage hybridization. Significantly, some of the hybridization events we identified involved human infective and non-human infective trypanosomes circulating in the same geographic areas.

## Author summary

Sexual reproduction allows genes from different individuals to be mixed up in the off-spring. This is particularly important for disease-causing microbes, because new

repository: DOI https://doi.org/10.5061/dryad.xd2547djb and NCBI Sequence Read Archive (SRA https://www.ncbi.nlm.nih.gov/sra) Project number: PRJNA795331.

**Funding:** We are grateful to the UK Biotechnology and Biological Sciences Research Council (https://bbsrc.ukri.org/) for funding (BB/R016437/1 to WG and TAW; BB/R010188/1 to WG). TAW is supported by a Royal Society University Research Fellowship (URF\R\201024 https://royalsociety.org/). The funders had no role in study design, data collection and analysis, decision to publish, or preparation of the manuscript.

**Competing interests:** The authors have declared that no competing interests exist.

combinations of harmful traits can arise, potentially leading to more severe outbreaks of disease. Tsetse-transmitted trypanosomes are single-celled parasites that cause severe human and livestock diseases in tropical Africa. During their developmental cycle in the tsetse fly, trypanosomes can mate and produce hybrid trypanosomes, which have one set of chromosomes from each parent. But polyploid hybrids, with more than one set of chromosomes from one or both parents, are often observed too. Here we have investigated how these polyploid hybrids are formed by comparing the genomes of hybrid progeny with those of their parents. Analysis of the large, paired chromosomes of both diploid and polyploid hybrids showed frequent crossovers, which are the hallmark of meiosis, the special form of division that produces haploid gametes. This indicates that the polyploids were formed after meiosis rather than by fusion of the parental diploid cells. We also investigated the inheritance of two other features of trypanosomes: the large family of variant surface glycoprotein (VSG) genes, and the mitochondrial (kinetoplast) DNA. Hybrid clones had inherited about half the *VSG* genes from each parent, and also showed biparental inheritance of one component of the kinetoplast DNA, the minicircles. We assessed the relatedness of field-collected trypanosomes by comparing their *VSG* and minicircle repertoires, together with maxicircle genotype. While most isolates shared few *VSG*s or minicircles, a group of mostly human-infective strains from Uganda had a large proportion of their repertoires in common. Most of these trypanosomes were probably related by clonal descent, but we also identified that some were hybrids by the mismatch between their maxicircle genotype and their *VSG* and minicircle repertoires. These signals of hybridization were also detected in some of the other field-collected isolates, suggesting that genetic exchange is widespread in nature. Significantly, the hybridization events involved human infective and non-human infective trypanosomes circulating in the same geographic areas, providing a mechanism for the generation of new, potentially more pathogenic, trypanosome strains causing human disease.

## Introduction

Sexual reproduction allows the mixing of genes from different individuals with formation of hybrid progeny. For microbial pathogens this is particularly important, as new combinations of traits such as drug resistance or virulence may be generated, potentially leading to more pathogenic strains and outbreaks of disease. Tsetse-transmitted trypanosomes such as *Trypanosoma brucei* and *T. congolense* are parasitic protists that cause severe human and livestock diseases in tropical Africa. During their developmental cycles in the tsetse fly, these trypanosomes undergo complex cycles of differentiation and proliferation in the fly's alimentary tract, ending up as infective metacyclics that are transmitted to the next host via the fly's saliva. In addition, *T. brucei* undergoes sexual reproduction in the fly's salivary glands, involving meiosis and production of haploid gametes [1–4], though many details remain to be elucidated. Consistent with meiosis, inheritance appears to follow Mendelian rules according to microsatellite analysis and most hybrid clones are diploid like the parental trypanosomes [5,6], but hybrids with high DNA contents, interpreted as triploid or tetraploid, also occur with some frequency [2,5,7,8]. Limited analysis of triploid hybrids has demonstrated the presence of three copies per genome of some housekeeping genes [5,7], but no genome-wide analysis has been carried out to date. Studies of chromosome inheritance after sexual reproduction in other members of the trypanosomatid family have shown that hybrid progeny are often polyploid. For example, in an experimental cross of *T. cruzi*, the hybrid progeny appeared to be the products of fusion

of the diploid parental trypanosomes, though with subsequent genome erosion [9]. Polyploid hybrids are frequently found in experimental crosses of *Leishmania* spp. [10–14]; indeed, of 24 hybrid clones from an *in vitro* cross of *L. tropica*, 19 (79%) were 3N or 4N [15].

In trypanosomes the mitochondrial DNA is tightly packaged into a unique organelle, the kinetoplast. The kinetoplast DNA (kDNA) consists of two sizes of circular DNA molecules: ~25 kb maxicircles, which encode genes required for mitochondrial function, and ~1 kb minicircles, which encode the guide RNAs used to edit maxicircle transcripts; maxi- and minicircles are intercalated into a single, giant network [16–18]. Analysis of the inheritance of kDNA in experimental crosses of both *T. brucei* and *Leishmania* spp. has revealed that hybrid clones have heterogeneous networks consisting of a mixture of parental minicircles; initially, the maxicircles are also heterogeneous, but after several generations of mitotic division and random partition between daughter cells, the relatively small number of maxicircles (~50) becomes homogeneous, so that inheritance of maxicircles appears to be uniparental [19–22]. These observations led to the hypothesis that the parental kDNA networks blend in the zygote, implying that both mitochondrial and cell fusion occur. However, the mechanism remains unknown and the very idea seems inconceivable to some, considering the elegant and highly controlled replication of kDNA [23].

Notwithstanding the experimental results from the laboratory, it has long been controversial how much sexual reproduction influences the population dynamics of *T. brucei* in nature [24–26]. This is epidemiologically important as both human-infective subspecies, *T. b. gambiense* (*Tbg*) and *T. b. rhodesiense* (*Tbr*), mingle with the non-human-infective subspecies *T. b. brucei* (*Tbb*) in infected mammalian and tsetse hosts, potentially facilitating mating and the generation of new strains of human-infective parasites. For example, transfer of the Serum Resistance Associated (SRA) gene from *Tbr* to *Tbb* generates new strains of the human infective parasite [27] and microsatellite analysis strongly supports the hypothesis that admixture between *Tbr* and *Tbb* has occurred in the past [28]. In contrast, the major group of *T. b. gambiense*, *Tbg1*, appears clonal, backed by comparison of whole genomes [29].

Here we have compared genomes of parental and hybrid progeny from a *T. brucei* cross, which produced presumed triploid and tetraploid progeny as well as the expected diploids (**Table 1**) [2], with the aim of verifying ploidy and deducing the mechanism of polyploidization. We confirmed the biparental inheritance of kDNA minicircles and of the variant surface glycoprotein (VSG) gene repertoires in hybrid progeny. In contrast to the parental trypanosome strains, hybrid progeny shared similar minicircle and *VSG* repertoires, and we searched for these signatures of hybridization among a collection of field-derived isolates.

**Table 1. Attributes of parental and hybrid clones.**

| Trypanosome clone | Fluorescence colour | DNA content | Maxicircle type | Microsatellite alleles | | |
|---|---|---|---|---|---|---|
| | | | | PLC | XI-53 | III-2 |
| J10 RFP | R | 2C | J10 | ab | aa | ab |
| 1738 GFP | G | 2C | 1738 | cd | bc | cd |
| FIG2 (SG22 clone 7) | G | 2C | J10 | bc | ac | ad |
| FIR1 (SG22 clone16) | R | 2C | mix | bc | ac | bc |
| F1R3N (SG1 clone 18) | R | 3C | 1738 | ad | ac | bd? |
| F1Y4N (SG1 clone 4) | Y | 4C | J10 | bd | ab | ad |

Fluorescence colour: R = red, G = green, Y = yellow i.e. red and green. Data from [2].

## Results

### Genome assembly

Genome data was obtained from four hybrid clones originating from an experimental cross of *T. brucei* J10 and 1738 [2] (**Table 1**). Previous measurements of DNA content showed that two of the hybrid clones (F1R1 and F1G2) were diploid like the parental trypanosomes, while two had high DNA contents consistent with triploidy (F1R3N) or tetraploidy (F1Y4N) [2]. Genome data from the parental lines J10 and 1738 was kindly provided by Adalgisa Caccone (Yale University) [30]. Details of isolates and assembled genomes can be found in **S1 Table**. Assembly of the hybrid isolates yielded similarly sized genomes from 150 bp read data with good contiguity (44.1–46.4 Mbp; N50 length 4959–5590 bp). Parent and field isolates sequenced from 75 bp read data showed more variation, with differences in total assembled size and contiguity likely influenced by the differences in read length and quality of sequencing. The assembled parent genomes were smaller (1738, 33.7 Mbp; J10, 36.9 Mbp), but had good contiguity (N50 length: 1738 13,768 bp, J10 8806 bp).

### K-mer and SNP analysis of heterozygosity

We used k-mer frequency analysis of the unassembled reads as an initial guide to heterozygosity, genome complexity and coverage depth [31,32]. This analysis calculates the frequency of unique nucleotide sequences (k-mers) in the genomic reads. Thus, diploid cells will have a k-mer frequency plot with two peaks representing heterozygous and homozygous loci (1x and 2x coverage depth respectively); additionally, in the *T. brucei* genome the 1x peak will also include monoallelic *VSG* genes. The assembled *T. brucei* Lister 427 genome [33] was used as a reference to align unassembled reads for the identification of heterozygous SNPs in the core regions of the chromosomes excluding subtelomeric *VSG*s. In principle, any unique reference position in a diploid genome will agree, disagree, or have an observed allele frequency half way between the two (= heterozygous). The number of observed heterozygous sites per genome is a measure of heterozygosity and changes in ploidy would alter the observed allele frequency.

**Fig 1** shows the k-mer and SNP frequency plots for the parent and hybrid trypanosomes. As expected, 1738 and J10 have two k-mer peaks, with J10 apparently having greater heterozygosity than 1738, as it has more heterozygous sites in the SNP plot (**Fig 1A and 1B**). Likewise, the diploid hybrid clone F1G2 has two peaks, with a prominent 1x peak (**Fig 1C**); the SNP frequency plot shows a far greater number of heterozygous SNPs than either parent, showing a gain in heterozygosity. Surprisingly, the other diploid hybrid clone (F1R1) did not conform to this pattern, with both the k-mer and SNP frequency plots showing multiple peaks (**Fig 1D**), suggesting that this "clone" consisted of a mixture of trypanosome strains. However, none of our subsequent analyses (see below) revealed the presence of nuclear or kinetoplast DNA from any strain outside the J10 x 1738 cross, and a mixture of two different hybrid clones would have produced anomalously large repertoires of *VSG* genes and kDNA minicircles. Therefore, our working hypothesis is that only genetic material from F1R1 is present, but that it has been reorganised through selfing. Of note is that the DNA used for genome sequencing here was derived after fly transmission of the original F1R1 clone without re-cloning, on the assumption that clones are transmitted faithfully because selfing only happens infrequently [34–36]. The complexity of the k-mer and SNP frequency plots (**Fig 1D**) suggests that a mixture of F1R1 and a selfed population is present. However, mixtures of F1R1 with a diploid selfed population would give rise to SNP peaks at approximately 0.25, 0.5 and 0.75, whereas the observed peaks are at approximately 0.2, 0.4, 0.6 and 0.8 (**Fig 1D**); in particular, the lack of the 0.5 peak does not fit with any mixture containing a diploid selfed population, suggesting instead that a

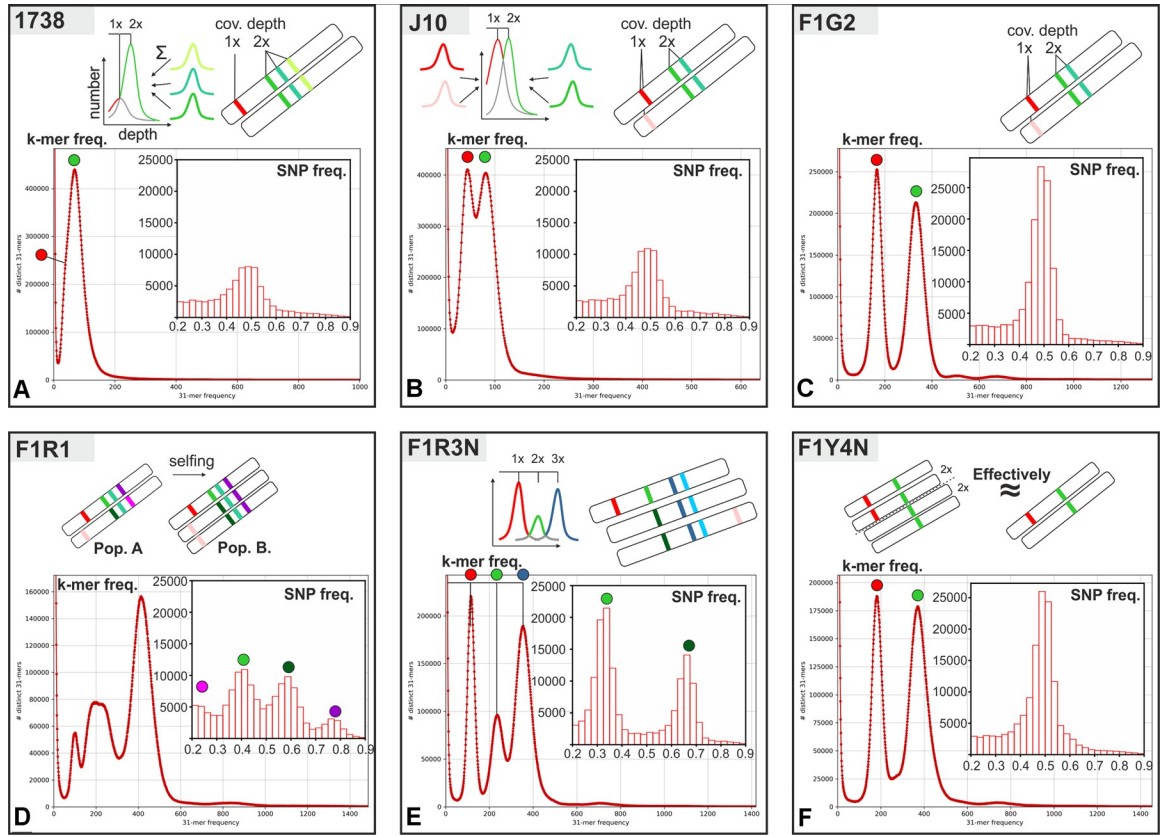

**Fig 1. K-mer and SNP analyses reveal ploidy and heterozygosity of hybrid clones.** Unassembled reads from parents (1738 and J10) and hybrid progeny (F1G2, F1R1, F1R3N, F1Y4N) were analysed by k-mer and SNP frequency; the plots are linked schematically to coverage depth via a genome model in the inset cartoons. The k-mer frequency plots show the number of k-mers with a specific coverage depth; equally spaced peaks correspond to multiples of k-mer incidence within the whole genome. The SNP frequency plots show the number of detected heterozygous SNPs within the core chromosomal regions, excluding *VSG*s in subtelomeric regions, and their relative observed allele frequency (proportion of observed reads with alternate value). Inset cartoons illustrate the inheritance of loci on one pair of homologues, linked by coloured dots to particular peaks on the frequency plots. Our working hypothesis is that F1R1 is a mixture of the original F1R1 clone (population A) and a selfed, possibly triploid, population (population B) formed during fly transmission of F1R1 after its original isolation; the 0.2:0.8 peak ratio is produced by chromosomal crossing over (see text for further explanation).

triploid selfed population is present, and interestingly this hypothesis is supported by the Smudgeplot [37] ploidy analysis (**S1 Fig**).

Despite the uncertainties in interpretation, we kept F1R1 in the analysis, because the original hybrid is present and makes up a substantial portion of the population; no information about its genomic repertoire has been lost and the presence of a selfed population only changes the observed allele frequency/coverage depth. Furthermore, this is an example of an interesting biological event and demonstrates the loss of heterozygosity on selfing.

For the two hybrid clones with raised DNA contents, the k-mer frequency plot for F1R3N has three peaks, and the SNP frequency plot has observed allele frequencies approaching 0.33 and 0.66 (**Fig 1E**), in accord with results generated for a triploid *T. congolense* isolate [38], while F1Y4N is indistinguishable from F1G2 (**Fig 1F**). F1Y4N had a DNA content consistent with tetraploidy, but microsatellite analysis detected only two alleles at each locus (**Table 1**), suggesting genome endoreplication rather than fusion of four independent gametes [2]; this observation is supported by the observation of only a single peak at 0.5 in the SNP frequency

analysis. To support our overall interpretation of ploidy in the hybrid clones, we carried out complementary analysis using Smudgeplot [37], included as a supplementary figure (**S1 Fig**).

## Inheritance of parental chromosomes by hybrid progeny

SNP analysis was further used to identify the two parental homologues for each of the 11 megabase chromosomes, which carry housekeeping genes [39], and track their inheritance in the four hybrid progeny clones (**Figs 2** and **S2**). As expected, the pattern of inheritance in the diploid progeny clone F1G2 is Mendelian, as there is one homologue from each parent and evidence of at least one crossover event for several chromosomes; the pattern for the original F1R1 hybrid was able to be resolved and is similar, supporting the interpretation from the k-mer analysis (**Fig 1D**) that this "clone" is a mixture of F1R1 and a minor selfed population, rather than another trypanosome strain or hybrid clone. The tetraploid clone F1Y4N follows the same pattern, as does the triploid clone F1R3N for chromosomes inherited from parent 1738 but not from parent J10 (**Fig 2**). Instead SNPs from both J10 homologues are present over extensive regions of seven of the 11 chromosomes (1, 4, 6, 7, 9, 10, 11), demonstrating that both homologues have been inherited and that this happened after meiotic crossing over had occurred.

The way the 3N hybrid was formed can be inferred from these observations. Firstly, the presence of a single 1738 homologue for each of the 11 chromosomes in F1R3N, nine of which have crossovers, indicates the contribution of a haploid 1738 genome that was a product of meiosis. The J10 partner also contributed a post-meiotic genome, as there is evidence of

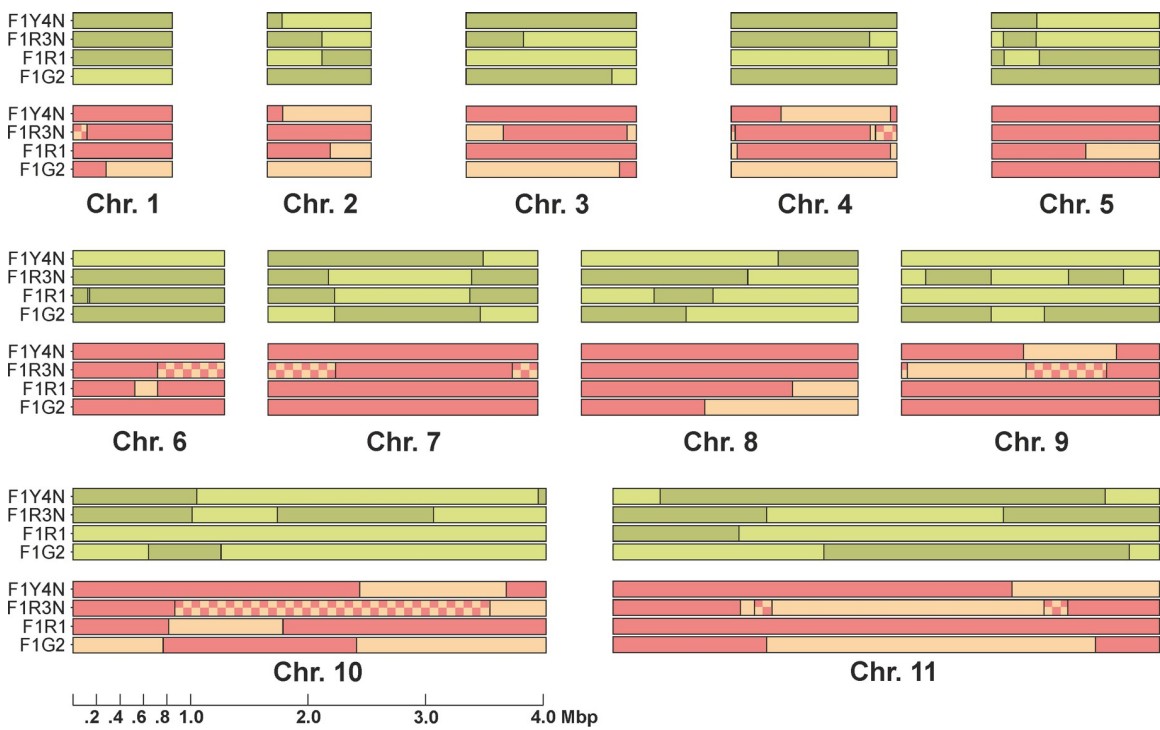

**Fig 2. Pattern of inheritance of parental SNPs in four hybrid progeny clones.** The core regions of the 11 chromosomes containing housekeeping genes are shown to scale. The two homologues from parent *T. b. brucei* 1738 are depicted in two shades of green, while those from J10 are red/orange. There is evidence of at least one crossover for most chromosomes as portions of both parental homologues are present in the chromosomes of the hybrids. The chequered blocks show regions where SNPs from both parental homologues were present. More detailed introgression maps are shown in **S2 Fig**.

crossing over for eight of the 11 chromosomes, which rules out the possibility that a vegetative diploid J10 cell fused with a haploid 1738 gamete. The fact that at least seven of the 11 chromosomes are present in two copies makes it unlikely that multiple, independent non-disjunction events occurred during meiosis in the J10 parent. Instead, the probable scenario is that the 1738 gamete fused either with a J10 cell in which meiosis was incomplete, or with two independent J10 gametes. We recently proposed a model of trypanosome meiosis in which gametes are produced sequentially and some intermediate stages have multiple nuclei [40], hence it is easy to envisage a scenario where fusion of a 1738 gamete with a J10 meiotic intermediate would produce the triploid genome observed in hybrid F1R3N. Moreover, we showed that meiotic intermediates, as well as gametes, expressed the membrane fusogenic protein HAP2 [40], potentially increasing the likelihood that meiotic intermediates, as well as gametes, fuse. Fusion of more than two gametes is also a possibility, although our observations of live gametes have mostly been of interacting pairs [3,41].

For the tetraploid clone F1Y4N, the chromosomal SNP patterns are indistinguishable from the diploid clone F1G2. This would not be the case if two diploid vegetative cells had fused, as both homologues from each parent would be present in the hybrid, and there would be no evidence of crossovers. The possibility that four independent haploid gametes fused is also ruled out by the absence of any regions of overlap (chequered blocks) in all 11 chromosomes (**Fig 1**). Fusion of meiotic intermediates is also ruled out, because again this would yield regions of overlap and moreover would need to have occurred in both parents. The remaining possibility is that endoreplication of parental chromosomes occurred after zygote formation, a parsimonious explanation since chromosomes from both parents would be duplicated in a single event.

## Inheritance of VSG repertoire in hybrid clones

A total of 3120 *VSG* genes and fragments were identified from the assembled parental and hybrid genomes. Of these, 2256 greater than 200 amino acids in length were clustered with CD-HIT [42] to identify *VSG*s with >99.5% sequence identity; 617 clusters were identified together with 296 singletons that were present in only one genome. Considerably more *VSG* genes were recovered from the genomes of the hybrid clones than the parents (**Fig 3**), reflecting the higher quality of genome data obtained from the hybrid clones compared to the parents, coupled with our stringent search criteria for identifying *VSG* genes. The total number of *VSG*s per genome found here falls far short of the typical 1000 *VSG* estimate derived from DNA hybridization densitometry or genome assembly [39,43], since these totals include closely similar genes, which would be clustered here, and pseudogenes, whereas only open reading frames were counted here. Overall 573 *VSG*s were recovered from the parental genomes, but only 14 *VSG*s were found in both parents, most of the *VSG* repertoire being strain-specific. This allowed the parental origin of individual *VSG*s to be assigned in the hybrid progeny clones (**Fig 3**). Each clone had inherited approximately half its *VSG*s from each parent, with a bias to inheritance from J10 (paired t test, p = 0.0335), perhaps because J10 had more *VSG*s than 1738, or that the assembly of 1738 was less contiguous/complete.

Some *VSG* genes were found together on contigs as *VSG* gene arrays, while others occurred as isolated genes. Contigs containing more than two *VSG* genes were identified in hybrid progeny only, reflecting the higher quality of these genomes. There were a total of 41 arrays of 3–8 *VSG* genes, which probably represent the subtelomeric *VSG* arrays found on each arm of the 11 diploid chromosomes [33,39], but may also originate from smaller chromosomes. Considering the parental origin of these 41 arrays, there is a skew towards inheritance from J10: 25 were unambiguously from J10 and 12 from 1738, with 4 unassigned. Inheritance of these arrays in individual hybrid clones follows the same pattern, with roughly twice as many arrays

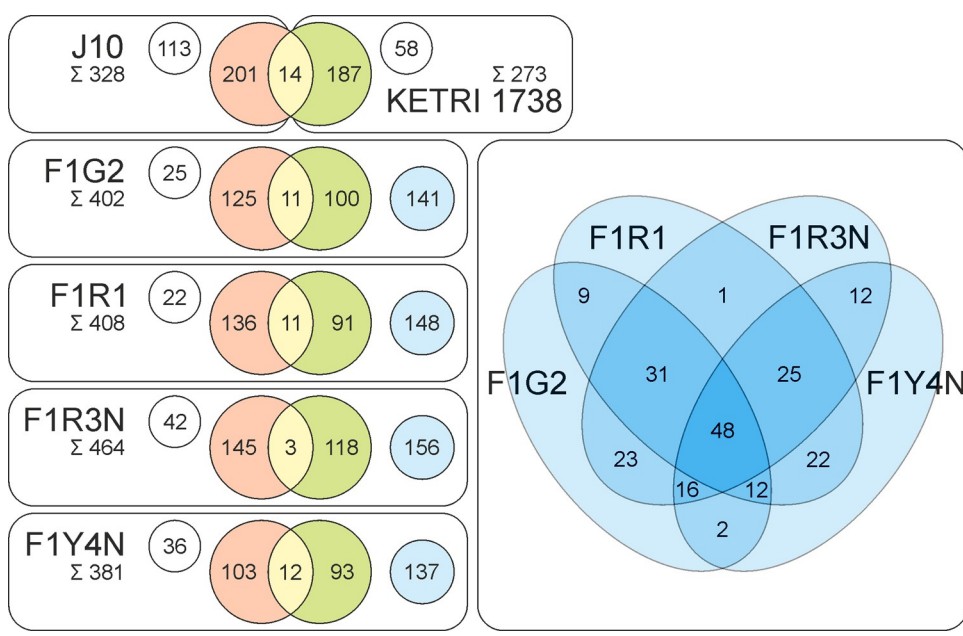

**Fig 3. Inheritance of parental *VSGs* in hybrid progeny clones.** Top: 587 *VSGs* were found in parents J10 (328 *VSGs*) and 1738 (273 *VSGs*) with 14 shared; those also found in one or more progeny clones are in red and green respectively, while strain-specific *VSGs* not found in progeny clones are shown in white circles. Bottom: *VSGs* inherited by the four hybrid progeny clones; parental origin is indicated by red (J10) and green (1738). Additional *VSGs* not found in either parent are shown in white circles if clone-specific, or blue circles if shared by one or more hybrid clones; the Venn diagram (right) expands the blue circle total to show the number of *VSGs* shared between individual hybrid clones.

from J10 as 1738 (mean no. of arrays from J10 and 1738 respectively = 17.50 and 6.75; paired t test, p = 0.0074; **Table 2**). This was true for diploid (F1G2) as well as polyploid (F1R3N, F1Y4N) hybrid clones, and does not therefore result from inheritance of extra J10 chromosomes.

## Inheritance of metacyclic VSG expression sites

Metacyclic *VSG* genes (*MVSGs*) are found in short expression sites (ES) at the ends of chromosomes and are distinguished from bloodstream form expression sites (BESs) by the lack of ES-associated genes (*ESAGs*) and the presence of an upstream *MVSG* promotor [44–49]. Screening genomic contigs for the *MVSG* promotor produced 47 contigs, which could be resolved to 13 clusters with sequence identity >99.5%. The longest contig in each cluster terminated downstream of the promoter and *VSG* in telomeric repeats (TTAGGG) (with the exception of cluster 6), with any *ESAGs* identified located upstream of the promoter, giving confidence that these loci represent genuine metacyclic expression sites (MES; [44–49]). Expression of these *MVSGs* has also been observed in transcriptomic analysis of individual trypanosomes isolated

**Table 2. Comparison of numbers of *VSG* gene arrays inherited by hybrid progeny from parents J10 and 1738.**

| Trypanosome clone | DNA content | Origin of *VSG* gene array | | | Total arrays |
|---|---|---|---|---|---|
| | | J10 | 1738 | Unassigned | |
| F1G2 | 2C | 18 | 5 | 3 | 26 |
| F1R1 | 2C | 20 | 6 | 4 | 30 |
| F1R3N | 3C | 19 | 10 | 1 | 30 |
| F1Y4N | 4C | 13 | 6 | 4 | 23 |

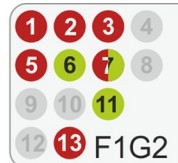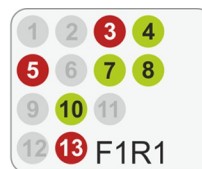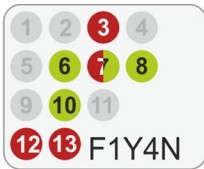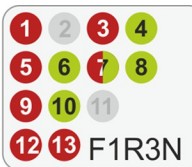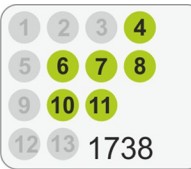

**Fig 4. Metacyclic *VSG* expression sites in parents and hybrid progeny.** Contigs containing *mVSG* promoters were clustered by neighbour-joining into 13 distinct loci, represented by coloured circles, where parental origin is indicated by colour (red = J10, green = 1738). The parents J10 and 1738 share only locus 7 in common (within 4% identity over the aligned region). Each hybrid has inherited loci from each parent as shown. For locus 7, both parental copies were present in F1G2, F1Y4N and F1R3N. F1R3N has all but two of the parental loci.

from the tsetse salivary glands [50]. Eight MES were found in J10 and six in 1738, with cluster 7 shared between them (~96% sequence identity over aligned region). All putative MES identified in the hybrid progeny were unequivocally of parental origin, with a mixture inherited from each parent (**Fig 4**); while most hybrid progeny inherited 7–9 MES, the triploid F1R3N had 12 MES, suggesting increased diversity of its subtelomeric ends.

## Inheritance of kinetoplast DNA in hybrid clones

The kinetoplast DNA (kDNA) of *T. brucei* consists of ~25 kb maxicircles and ~1 kb minicircles. Entire maxicircle coding regions were obtained from the parental and hybrid genomes. Assembly of the parental maxicircles revealed 50 SNPs and 15 single base pair indels distinguishing the maxicircles of J10 and 1738. Reads were aligned to the assembled sequences and no heteroplasmy was observed. Alignment of the maxicircle coding regions from the four hybrid clones showed that F1G2 and F1Y4N had maxicircles identical to those of J10, while F1R1 and F1R3N had maxicircles identical to those of 1738; no additional variation or recombination was observed. These results confirm our earlier findings based on presence of a single *Hin*fI site in the cytochrome oxidase gene (**Table 1**), except that F1R1 was reported to have maxicircles of both parental types [2]. As this prior result was obtained during early passage of clone F1R1, we assume that maxicircles since became homogeneous through sequential vegetative divisions [21,51]. Again, the presence of only one maxicircle genotype in F1R1 confirms that it is not a mixture of two different strains, but a mixture of the original clone and a selfed population.

Sequences of 308 unique minicircles were recovered from the parental and hybrid genomes. Of these, 117 were found in parent J10 and 98 in parent 1738, while the hybrid clones had 213 (F1G2), 230 (F1R1), 219 (F1R3N) and 130 (F1Y4N) (**Table 3**). We assume that more minicircles were recovered from the hybrid genomes due to the higher quality of sequence data. The hybrid clones had inherited approximately equal numbers of minicircles from each parent (mean no. of minicircles from J10 and 1738 was respectively 72 and 62; paired t test, p = 0.0668; **Table 3**). A total of 67 minicircles were shared across the four hybrid clones, of which 28 were found in J10, 19 in 1738 and 20 in neither parent. It is noteworthy that the triploid and tetraploid hybrids had no excess of minicircles compared to the diploid hybrids.

**Table 3. Parental origin of minicircles in hybrid clones.**

| Parental origin of minicircle | F1G2 | F1R1 | F1R3N | F1Y4N |
|---|---|---|---|---|
| 1738 | 59 | 76 | 73 | 40 |
| J10 | 80 | 83 | 78 | 48 |
| Subtotal | 139 | 159 | 151 | 88 |
| Unassigned | 74 | 71 | 68 | 42 |
| Total | 213 | 230 | 219 | 130 |

In summary, we have confirmed uniparental inheritance of the maxicircle component of kDNA in hybrid progeny, but biparental inheritance of the minicircles in *T. brucei* [19–21].

## Comparison of field isolates

Comparison of the inheritance of nuclear and kinetoplast DNAs in laboratory crosses has shown that hybrids inherit a mixture of *VSG*s and minicircles from both parents, but inherit their maxicircle type from just one parent. Therefore, we investigated whether such patterns of inheritance could be identified among field-collected isolates to detect signatures of genetic exchange. Genome data from 33 isolates of *T. brucei* (16 *T. b. brucei*, 14 *T. b. rhodesiense*, 3 *T. b. gambiense*) and four isolates of *T. evansi* was kindly provided by Adalgisa Caccone (Yale University) [30]; details of isolates and assembled genomes can be found in **S1 Table** with k-mer and SNP analysis shown in **S3 Fig**. *VSG* genes and minicircles were recovered from each genome. For the *T. brucei* isolates, kinetoplast DNA maxicircle coding regions obtained previously [52] were used to derive a phylogenetic tree (**Fig 5A**). While the majority of maxicircle sequences were similar and formed a Pan-African group, three discrete clades were evident: Sindo, Kiboko and Lister 427 group. These groupings were used to order *T. brucei* isolates in the data matrix (**Fig 5B**), which displays the numbers of shared *VSG*s and minicircles between each pair of isolates; only *VSG* data is shown for *T. evansi*, because minicircles are largely homogeneous in this species [53].

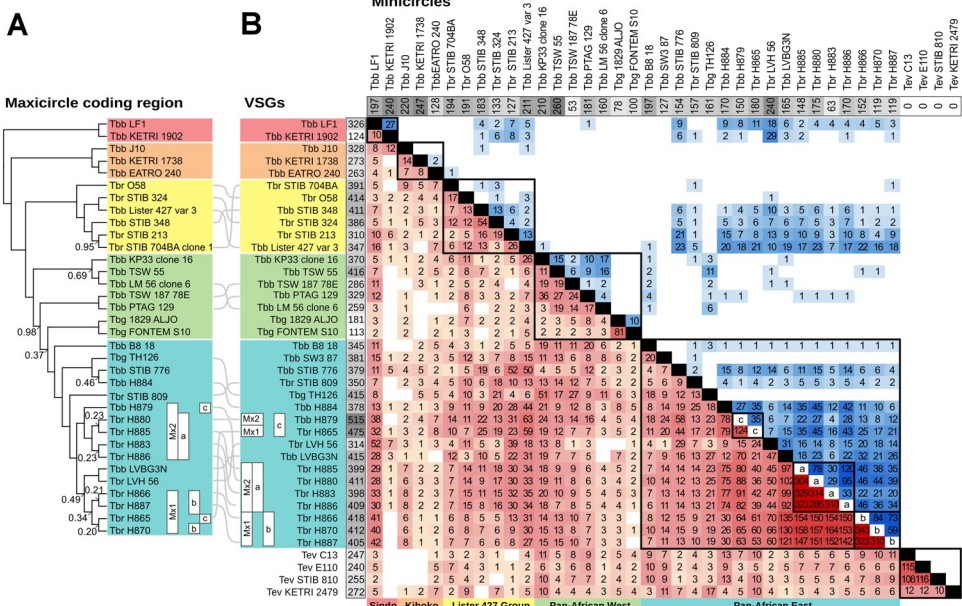

**Fig 5. Shared *VSG*s and kinetoplast DNA minicircles among field-collected isolates of subgenus *Trypanozoon*.** A. Phylogenetic tree of the maxicircle coding region of *T. brucei* isolates, excluding SW3/87, which produced a fragmented maxicircle on assembly. Node values show posterior probabilities <1. Isolates are grouped by colour: red, Sindo; orange, Kiboko; yellow, Lister 427 group; green, Pan-African West; cyan, Pan-African East. In this last group, two clades are evident among Ugandan isolates (Mx1 and 2), and the *VSG* cluster (a, b, c) is also shown. B. Data matrices of shared *VSG*s (red) and minicircles (blue) from 39 isolates of subgenus *Trypanozoon*. Higher values are indicated by darker colour. The grey boxes in the horizontal row and vertical column show total numbers of minicircles and *VSG*s recovered, respectively. Isolates are grouped by colour largely as in A, with SW3/87 now included in the cyan group and other small changes within group to highlight shared *VSG*s. As *T. evansi* isolates have only one major minicircle type, minicircle data is voided. For the Ugandan isolates, the *VSG* cluster (a, b, c) is indicated on the diagonal.

About a quarter of isolates, including J10 and 1738, had fewer than 25 *VSG*s in common with any other isolate (KETRI 1902, J10, 1738, EATRO 240, 058, STIB 704, LM 56, B8/18, SW3, KETRI 2479; **Fig 5B**), reflecting the immense diversity of *VSG* repertoires among trypanosomes separated by vast distances and collected at different times. Although isolates of *T. b. gambiense* Type 1 (*Tbg*1) and *T. evansi* Type A (*Te*A) were also collected separately in space and time, *Tbg*1 Aljo and Fontem shared 81 *VSG*s, and *Te*A C13, E110 and STIB 810 shared 108–116 *VSG*s, despite originating from three different continents. However, the *Te*A isolates shared only 12–14 *VSG*s with KETRI 2479, a Type B *T. evansi*, confirming the distinct nature of this subgroup [54–57].

The most overlapping *VSG* repertoires were found among *T. b. brucei* (*Tbb*) and *T. b. rhodesiense* (*Tbr*) isolates originating from the HAT focus in southeast Uganda, collected over a 20 year period from 1990 to 2010. Some of these isolates shared >300 *VSG* genes, a large part of their identified *VSG* repertoire, indicating that they are very closely related by descent, either as clonal lineages or through interbreeding. Among the ten isolates analysed here, three *VSG* clusters can be distinguished: (a) H880, H883, H885, H886, sharing 285–329 *VSG*s; (b) H866, H870, H887, sharing 310–343 *VSG*s; (c) H865, H879, sharing 124 *VSG*s (**Fig 5B**). H884, a bovine *Tbb* isolate from 2003, is an outlier, sharing 74–79 *VSG*s with isolates in clusters a and c, and only 26–30 *VSG*s with isolates in cluster b. Genomes of these ten isolates have been analysed previously by SNP analysis excluding *VSG*s [30] and clusters a–c correspond to clusters 5, 6 and 9 respectively, with the outlier H884 also placed in cluster 5. Similar relationships were evident from phylogenetic analysis of the SNP data, with isolates in clusters a–c distributed in three clades and H884 placed with cluster a isolates [58]. Concordance between the *VSG* and SNP data is expected, as both datasets relate to information on the 11 pairs of large chromosomes, though some *VSG*s are also found on smaller chromosomes.

Minicircles were rarely found in common between isolates, even those that shared maxicircle type (**Fig 5B**); for example, the three Kiboko isolates (J10, 1738, EATRO 240) shared 0–2 minicircles, *Tbg*1 isolates Aljo and Fontem shared only 10 minicircles, and the Lister 427 group shared a maximum of 13 minicircles. The largest numbers of shared minicircles occurred among the Ugandan isolates that also had extensively overlapping *VSG* repertoires. The question is whether these shared *VSG* and minicircle repertoires signify genetic exchange or simply result from clonal descent.

## Evidence for genetic exchange among Ugandan isolates

The ten Ugandan isolates all originate from SE Uganda, an epidemic that started in the 1970s and spread northwards [59–63]. Clusters a and b contain *Tbr* isolates collected in 2003–2010 and 1990–1992 respectively (**Table 4**) and represent two different *Tbr* genotypes circulating in the epidemic. A third *Tbr* genotype is represented by H865, isolated from a patient in 1990. H865 was found in cluster c, together with a bovine *Tbb* isolate, H879; as *Tbb* H879 was isolated nearly two decades after *Tbr* H865, the shared *VSG* genes could indicate clonal descent of H879 from H865, with loss of the *SRA* gene, which confers human infectivity [64], or past genetic exchange between *Tbb* and *Tbr*.

How can we distinguish clonal descent from genetic exchange? During clonal descent, the kinetoplast DNA maxicircles and minicircles will be inherited conservatively, but, as we have seen from the laboratory cross, hybrids inherit maxicircles from either parent and minicircles from both parents. Hence hybrids are predicted to show discordance between their genetic relatedness based on maxicircle genotype compared to minicircles and *VSG*s. Maxicircle SNPs divide the Ugandan isolates into two clades Mx1 and Mx2 (**Fig 5A**), corresponding broadly to *VSG* clusters a and b, with the two isolates in cluster c subsumed into either Mx1 (H865) or

**Table 4. *VSG* and kinetoplast DNA profiles of ten Ugandan isolates.**

| Isolate | Subspecies | Host | Isolation year | *VSG* cluster | Maxicircle clade | Minicircles maximum share |
|---------|-----------|------|----------------|---------------|------------------|---------------------------|
| H865 | Tbr | H | 1990 | c | Mx1 | H880 45/180 (25%) |
| H866 | Tbr | H | 1990 | b | Mx1 | H886 46/152 (30%) |
| H870 | Tbr | H | 1990 | b | Mx1 | H866 84/119 (71%) |
| H879 | Tbb | Bv | 2009 | c | Mx2 | H865 35/150 (23%) |
| H880 | Tbr | H | 2003 | a | Mx2 | H886 95/175 (54%) |
| H883 | Tbr | Dog | 2005 | a | Mx2 | H886 33/63 (52%) |
| H884 | Tbb | Bv | 2003 | - | - | H880 45/170 (26%) |
| H885 | Tbr | H | 2010 | a | Mx2 | H886 120/148 (81%) |
| H886 | Tbr | H | 2010 | a | Mx2 | H885 120/170 (71%) |
| H887 | Tbr | H | 1992 | b | Mx1 | H866 73/119 (61%) |

Subspecies designation is based on presence of the *SRA* gene [57]; Tbb = *T. brucei brucei*, Tbr = *T. b. rhodesiense*. Host: H = human, Bv = bovine. *VSG* cluster: according to numbers of *VSG*s shared between isolates (see **Fig 5B**); H884 shared <80 VSGs with any of these isolates. Maxicircle clade: see **Fig 5A**. Minicircles maximum share: maximum proportion of identified minicircles shared with another isolate (see **Fig 5B**).

Mx2 (H879), though statistical support for the exact position of H865 in Mx1 is low. In effect, maxicircle genotype has partitioned the isolates according to date of isolation, with clade Mx1 corresponding to 1990–1992 and Mx2 to 2003–2010 (**Table 4**). Thus, H865 (isolated 1990) and H879 (isolated 2009) have the common maxicircle genotype circulating during their separate eras, but do not share the expected *VSG* repertoire, a for Mx2 or b for Mx1 (**Fig 5A**), indicating that both are products of genetic exchange. In contrast, isolates from clusters a and b share both *VSG* repertoires and maxicircle genotypes (**Table 4**), as well as SNP profiles [30,58], and are probably linked by clonal descent. For clusters a and b, this conclusion is reinforced by the minicircle inheritance results, which show that a large proportion of minicircles are shared as well as *VSG*s (**Table 4**). However, for the two isolates in cluster c (H865, H879), minicircle repertoires are more heterogeneous in origin and do not show concordance with maxicircle genotype, both signatures of kDNA hybridization.

In summary, the evidence suggests that *Tbr* H865 and *Tbb* H879 represent lineages that have undergone genetic exchange, based on discordance between maxicircle genotype and *VSG* and minicircle repertoires. We cannot rule out the possibility that genetic exchange has also occurred among cluster a and b isolates, but clonal descent adequately explains the congruence between their kinetoplast DNA and *VSG* repertoires.

## Evidence for genetic exchange among distantly-related lineages

Mating between distantly related trypanosomes will yield hybrid progeny with melded *VSG* and minicircle repertoires, but maxicircles of either parental type. As these related clonal lineages diverge, the proportion of shared *VSG*s and minicircles will gradually diminish, while the maxicircles will also diverge through accumulated mutations. Hence signatures of old hybridization events might still be present in apparently unrelated isolates.

**Fig 6** highlights the relationships between *VSG* repertoire and maxicircle genotype among the 39 diverse isolates of subgenus *Trypanozoon* studied here. Within maxicircle genotype, isolates that share a large proportion of their *VSG* repertoire are connected by broad blue ribbons: Ugandan clusters a, b and c; *Tbg*1 Aljo and Fontem; *Te*A C13, E110, STIB 810. But connections between isolates of different maxicircle genotype are also evident. Most remarkable is the linkage between Sindo *Tbb* LF1 and isolates with the Pan-African East maxicircle genotype from the same geographical region surrounding Lake Victoria. While LF1 shares only 10 *VSG*s with

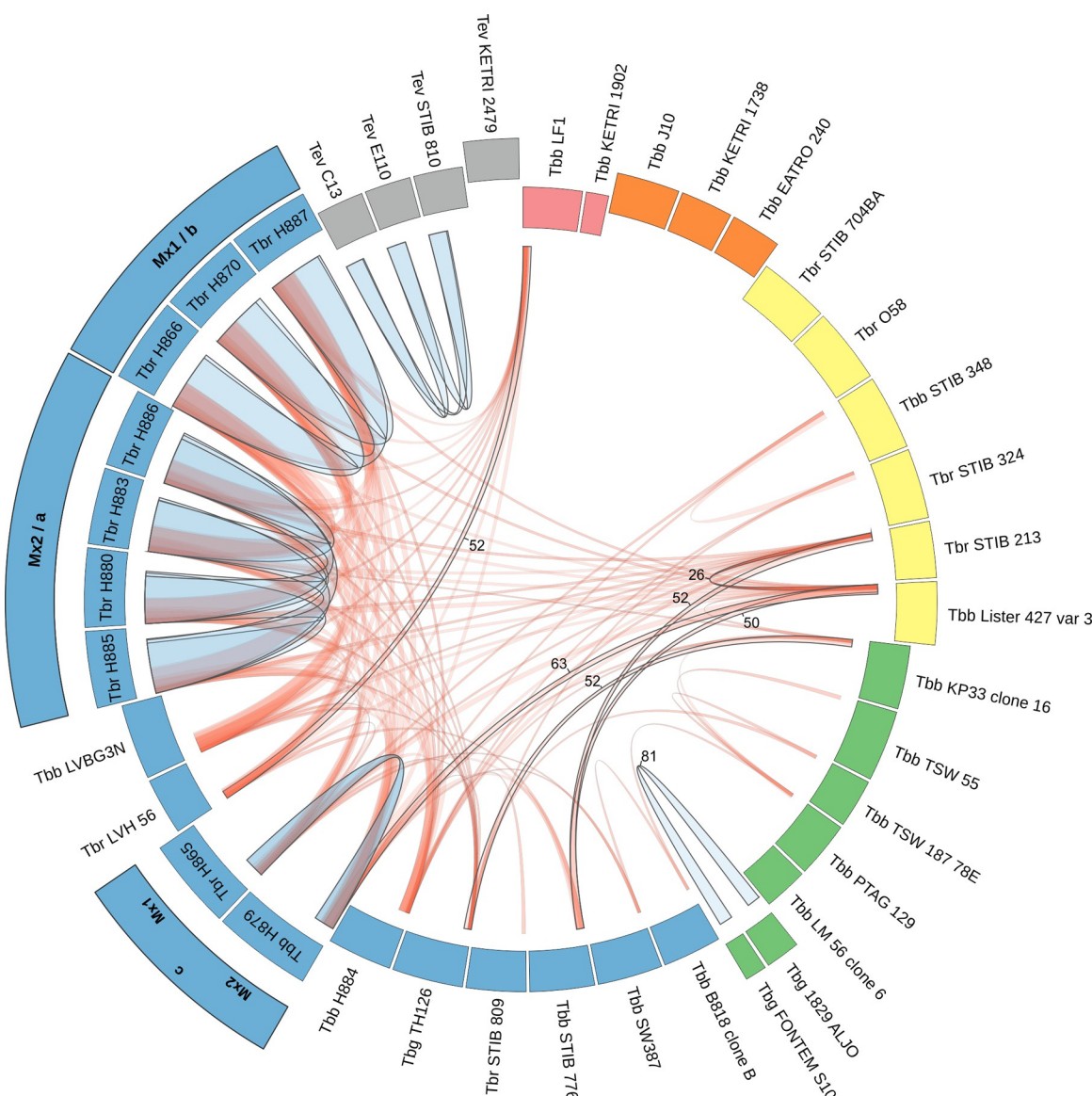

**Fig 6. Relationships between *VSG* repertoire and maxicircle type.** Data from the *VSG* matrix (**Fig 5B**) was filtered to show only linkages between isolates with more than 20 *VSG*s in common. The 35 *T. brucei* isolates are arranged and colour-coded according to the maxicircle phylogeny (**Fig 5A**), with the *T. evansi* isolates included as grey blocks. Maxicircle clade (Mx1 or 2) and *VSG* cluster (a, b, c) are shown for related Ugandan isolates. The block corresponding to each isolate, and the ribbons connecting them, are scaled to the number of *VSG*s; Ugandan isolates in *VSG* clusters a and b share the majority of their *VSG*s, as ribbons are almost as wide as the blocks. Ribbons are coloured red to help visualise more distant links, which are likely to reflect genetic exchange, from within-group links (blue); the numbers give the actual number of shared *VSG*s for selected links.

the other Sindo isolate, *Tbb* 1902, it shares 52 *VSG*s with *Tbr* LVH 56 from Lambwe Valley, Kenya, and 40–42 with Ugandan cluster b isolates (H866, H870, H887) (**Figs 5B** and **6**). Both LF1 and LVH 56, together with LVBG3N, originate from an early 1980's HAT outbreak in the Lambwe Valley [65], while *Tbb* 1902 was collected earlier (1971) from a waterbuck. Despite the divergence of maxicircle genotypes, the Sindo isolates LF1 and 1902 also share minicircles with LVH 56 (18 and 29 respectively, **Fig 5B**). Discordant maxicircle genotype combined with overlapping *VSG* and minicircle repertoires points to genetic exchange among trypanosomes in the Lambwe Valley, with further links to trypanosomes circulating in south-east Uganda.

The Lister 427 group also shows the hallmarks of genetic exchange, with strong linkages of 50 or more *VSG*s between STIB 213 and Lister 427 to Pan-African East isolates STIB 776, H879 and H865 (**Fig 6**); around 20 minicircles are also shared (**Fig 5B**).

In contrast, the Kiboko maxicircle genotype shows no linkages, either within the group (<14 *VSG*s in common) or to different maxicircle genotypes (**Fig 6**), and the majority of their identified *VSG*s (~80%) are unique to the isolate (**S2 Table**). As both parents in the experimental cross (J10 and 1738) belong to the Kiboko group, there is no question that isolates within this group are fully capable of mating, both within the Kiboko group [2] and with strains of different maxicircle genotype [20,66,67], demonstrating that this is not an asexual maxicircle clade. Analysis of more isolates, compared to the limited sampling in this study, may reveal stronger connections between maxicircle clades.

## Discussion

Comparison of the genomes of parents and hybrid progeny from an experimental cross of *T. brucei* has provided support for a number of hypotheses about the process of sexual reproduction in trypanosomes. There was already experimental evidence of a meiotic division in *T. brucei* [4] and SNP analysis has confirmed that crossovers occurred on all 11 pairs of megabase chromosomes when hybrid progeny were formed. SNP analysis also demonstrated the presence of two sets of post-meiotic chromosomes from one parent, together with one set from the other parent, in a hybrid with a 3C DNA content. This triploid hybrid was therefore most likely formed by fusion of a gamete with a meiotic intermediate rather than a vegetative diploid cell. Similarly, the demonstration of numerous crossovers in the chromosomes of a hybrid with a 4C DNA content showed that it did not arise by fusion of diploid cells, but probably by endoreplication of chromosomes post zygote formation.

Surprisingly, one of the two diploid hybrid progeny included in this analysis turned out not to be a pure clonal population, but a mixture of the original clone and a selfed population. This was evident from the k-mer analysis, but not suspected from inheritance of *VSG*s or minicircles. Tracing the derivation of the cryopreserved stock used for genome sequencing, revealed that it had been tsetse-transmitted without subsequent re-cloning. Our previous studies have shown that meiosis and production of haploid gametes occur in clonal populations during fly transmission [3,4], opening the possibility of selfing, though this is thought to be rare [34–36]. This assumption needs to be reconsidered in the light of the evidence of selfing revealed here.

Hybrid progeny inherited substantial numbers of *VSG* genes from both parents, demonstrating that sexual reproduction mixes up *VSG* genes and thereby generates novel *VSG* repertoires. This confirms previous results demonstrating immunologically that experimental hybrids had recombinant variable antigen type repertoires [68]. Besides the inheritance of chromosome-internal *VSG* gene arrays, we also showed that hybrid progeny inherit a mixture of metacyclic expression sites from both parents, each containing a metacyclic *VSG* gene; every hybrid is therefore equipped with a new combination of metacyclic *VSG*s, which may be a competitive advantage in successfully establishing infection in the mammalian host. In addition to exchange of nuclear DNA, trypanosomes recombine their mitochondrial (kinetoplast) DNA during mating. Though kinetoplast DNA (kDNA) maxicircles showed uniparental inheritance with no detectable evidence of heteroplasmy, the minicircles were inherited from both parents, giving rise to a hybrid kDNA network. Hybrid trypanosomes thus share minicircles with their siblings as well as their parents. **Fig 7** summarizes the different modes of inheritance of nuclear and mitochondrial genomes.

We used these observations to search for evidence of hybridization among trypanosome isolates from endemic areas. Isolates collected from geographically distant locations at

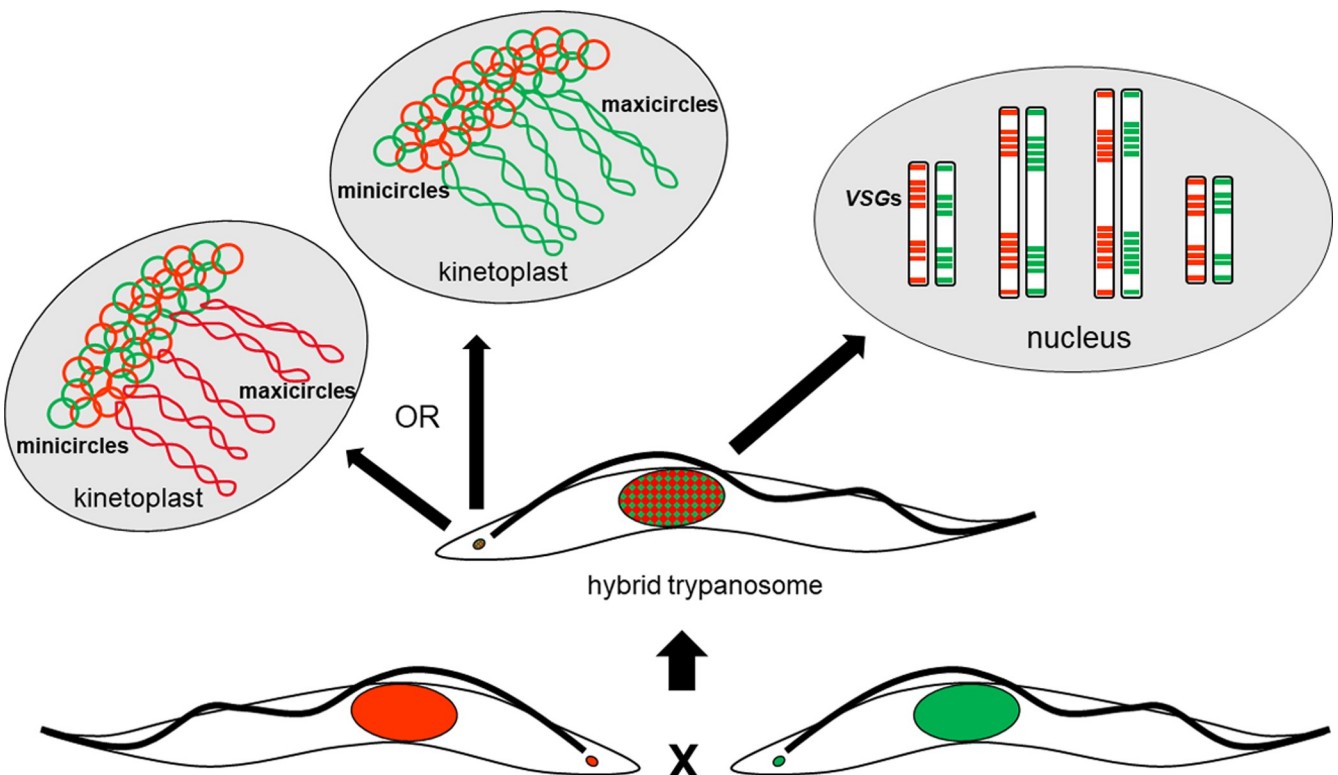

**Fig 7. Modes of inheritance of nuclear and kinetoplast DNA.** Diagram summarizing the different modes of inheritance of nuclear and mitochondrial genomes.

different times generally had non-overlapping *VSG* and kDNA minicircle repertoires, with some exceptions such as the clonally propagated genotypes *Tbg1* and *TeA*, reaffirming the consensus that these are widespread clonal lineages [28,57,58,69–72]. However, for nine Ugandan isolates, more closely associated in space and time, a large proportion of *VSG*s and minicircles were shared, demonstrating close genetic relatedness brought about by clonal descent or genetic exchange. While shared maxicircle genotype would be expected for clonal descent, divergent maxicircle and nuclear genotypes provided unequivocal evidence of hybridization. Just this limited sampling of isolates allowed us to build up a complex picture of interrelatedness, some by direct, clonal descent, and some by genetic exchange.

Across the broader range of isolates sampled, further instances of discordance between maxicircle genotype and *VSG* and minicircle repertoires were identified, suggesting widespread genetic exchange among *T. brucei* isolates and that many lineages are hybrid. This has resolved some previous discrepancies where isolates have been assigned to different phylogenetic groups according to maxicircle or nuclear DNA genotyping [73].

Significantly, there was evidence of hybridization between human infective and non-infective trypanosomes, providing a mechanism for the generation of new, potentially more pathogenic, strains of *Tbr*. *VSG* repertoires overlapped extensively among *Tbr* and *Tbb* isolates from East African HAT foci; for example, *VSG* cluster c from Uganda contains both *Tbr* and *Tbb* isolates with 124 *VSG*s in common, while *Tbb* LVBG3N from Lambwe Valley, Kenya, shared 121–135 *VSG*s with *Tbr* isolates from Uganda. Similarly in West Africa, *Tbg2* TH126 from the Daloa/Vavoua/Bouaflé HAT area in Côte d'Ivoire shared 52 *VSG*s with *Tbb* KP33 isolated from a tsetse fly.

## Methods

### Trypanosomes

Hybrid clones originated from an experimental cross of *T. b. brucei* J10 and 1738 [2] and comprised two diploid progeny F1R1 and F1G2 (originally called SG22 clone 16 and SG22 clone 7 respectively), which were subsequently used in F1 crosses [41], and two hybrid progeny with raised DNA contents presumed to be triploid (F1R3N = SG1 clone 18) or tetraploid (F1Y4N = SG1 clone 4; **Table 1**). The four hybrid clones were grown as procyclics in Cunningham's medium (CM) [74] supplemented with 10 μg/ml gentamicin, 5 μg/ml hemin and 15% v/v heat-inactivated foetal calf serum (FCS) at 27˚C. High molecular weight DNA for genome sequencing was purified from approximately $5 \times 10^8$ trypanosomes using a Blood and cell culture kit (Qiagen) and a modification of the manufacturer's yeast cell protocol. Briefly, cells were pelleted by centrifugation, washed once with PBS and resuspended in 5 ml lysis buffer containing proteinase and RNAase as per the manufacturer's protocol. Following 1 hour incubation at 50˚C, lysates were centrifuged at 5000 rpm for 5 minutes at room temperature in a microfuge to pellet debris before the supernatant was applied to a Genomic-tip 100/G column (Qiagen). Subsequent processing followed the manufacturer's protocol; after isopropanol precipitation, DNA was resuspended in 200 μl 10 mM Tris, 0.1 mM EDTA, pH 8 and stored at 4˚C.

### Genome sequencing and assembly

DNAs from the four hybrid clones (F1R1, F1G2, F1R3N, F1Y4N) were sequenced by the Earlham Institute, Norwich, UK, using Illumina NovaSeq with 150 bp paired end reads (data available from https://www.ncbi.nlm.nih.gov/sra Project no. PRJNA795331). Illumina sequence data (75 bp reads) from J10 and 1738, together with 37 other subgenus *Trypanozoon* isolates, was kindly provided by Adalgisa Caccone, Yale, USA [30]. Reads were assembled using SPAdes v3.13.1 [75]. K-mer frequency analysis was performed using Fastp [76], and histograms produced in KAT [32]. SNPs were identified by read alignment using BWA, processed with samtools and bcftools [77] before SNPs were called using Freebayes. RTG Tools and Tabix [78] were used to produce numerical reports and a Tablet v1.19.09.03 [79] and Artemis [80] were used to visualise SNP distribution. Further details of genomes and bioinformatic analyses are provided in **S1 Table** and **S1 Text**. The data are available from the Dryad repository: https://doi.org/10.5061/dryad.xd2547djb [81].

### *VSG* gene analysis

*VSG* open reading frames were predicted from the contig pool using Transdecoder [82] and *VSG*s identified by Phmmer [83] using the Pfam HMM for the *VSG* C-terminal domain. For cluster analysis, sequences of greater than 200 amino acids in length with a Phmmer score > 1e-6 were clustered with CD-HIT [42]. Clusters were defined by proteins with regions overlapping 95% of total length sharing 99.5% sequence identity. Further details of bioinformatic analysis are provided in **S1 Text** and the data are available from the Dryad repository: https://doi.org/10.5061/dryad.xd2547djb [81].

### Metacyclic expression sites

An HMM was derived from known metacyclic promoter sequences [47,49] and nhmmer [84] was then used to screen genomic contigs with this model. Sequences were clustered by alignment using MAFFT and tree construction via neighbour-joining. These groups were then confirmed by all vs all BLAST; the great variation in contig length and position of overlap made

this method preferable over CD-HIT, though largely similar results were obtained. Further details of bioinformatic analysis are provided in **S1 Text** and the data are available from the Dryad repository: https://doi.org/10.5061/dryad.xd2547djb [81].

### Kinetoplast DNA analysis

Minicircles were identified from the contig SPAdes pool using nhmmer [84] with HMM derived from *T. brucei* minicircle sequences from the public Entrez database, using an e-value threshold of 1e-6. Reads aligning to these sequences were extracted from the read pool using Magic-BLAST [85] and a specialised sub-assembly of the reads performed with SPAdes using a kmer value of 55 and the plasmid mode. Circular assembled molecules were filtered by size and HMM. Sequences were oriented uniformly by identifying the minicircle conserved sequence blocks by BLAST v2.2.31+ [86]. Clustering was performed using CD-HIT [42] with thresholds of >98% identity for sequences within 90% of total length. Maxicircles were identified from the contig pool using BLAST v2.2.31+ [86]. Sequences were oriented and aligned using MAFFT v7.427 [87] and trimmed to the coding region. Site differences were identified using SNP-sites [88]. Heteroplasmy was assessed by aligning the reads to assembled contigs using BWA MEM v0.7.17 [89] and visualising the alignment in Tablet v1.19.09.03 [79]. A phylogeny was produced from the alignment using IQ-Tree ModelFinder [84] to compare base substitution models and parameters, and BEAST [90] to perform the final phylogeny. Trees were sampled every 1000 iterations over a chain length of 10,000,000; run results were visualised in Tracer [91] and consensus tree by Treeannotator v1.10.4 [90]. Further details of bioinformatic analysis are provided in **S1 Text** and the data are available from the Dryad repository: https://doi.org/10.5061/dryad.xd2547djb [81].

### Dryad DOI

https://doi.org/10.5061/dryad.xd2547djb

## Supporting information

**S1 Fig. Smudgeplot analysis of hybrid and parent read data.** Read pools were quality filtered with Fastp and k-mer histogram tables were then analysed with Smudgeplot (Ranallo-Benavidez TR et al 2020 doi.org/10.1038/s41467-020-14998-3). Proposed ploidy is shown under isolate name, with the probability of other karyotypes shown on the right. Parental strains 1738 and J10, together with hybrid progeny clones F1G2 and F1Y4N, all fit to diploid, while F1R3N fits best to triploid. The anomalous hybrid clone F1R1 has an unusual intermediate pattern inconsistent with a pure diploid population.
(TIF)

**S2 Fig. Introgression maps of hybrid clones.** Introgression maps were constructed from patterns of inherited heterozygous SNPs from the parental strains. There were ~20,000 SNPs in each hybrid and the figure illustrates their density and distribution.
(TIF)

**S3 Fig. K-mer analysis of genomic reads from *T. brucei* and *T. evansi* isolates.** K-mer analysis for the 37 additional field isolates. Although k-mer peaks are unresolved for some isolates, the range of peak shapes suggests that these populations have varying levels of heterozygosity.
(TIF)

**S1 Table. Details of trypanosome isolates and assembled genomes.**
(XLSX)

**S2 Table. *VSG* repertoire by proportion shared by maxicircle clade.**
(TIF)

**S1 Text. Supplementary bioinformatics methods and data.**
(DOCX)

## Acknowledgments

Many thanks to Adalgisa Caccone for supplying raw read sequence data from field isolates.

## Author Contributions

**Conceptualization:** Christopher Kay, Tom A. Williams, Wendy Gibson.

**Data curation:** Christopher Kay.

**Formal analysis:** Christopher Kay, Lori Peacock.

**Funding acquisition:** Tom A. Williams, Wendy Gibson.

**Investigation:** Christopher Kay, Lori Peacock.

**Methodology:** Christopher Kay, Tom A. Williams.

**Project administration:** Tom A. Williams, Wendy Gibson.

**Resources:** Lori Peacock, Wendy Gibson.

**Supervision:** Tom A. Williams, Wendy Gibson.

**Visualization:** Christopher Kay, Wendy Gibson.

**Writing – original draft:** Wendy Gibson.

**Writing – review & editing:** Christopher Kay, Lori Peacock, Tom A. Williams.

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
