## [Decision Letter · Decision Letter 0]

13 Sep 2021

Dear Professor Gibson,

Thank you very much for submitting your manuscript "Signatures of hybridization in Trypanosoma brucei" for consideration at PLOS Pathogens. As with all papers reviewed by the journal, your manuscript was reviewed by members of the editorial board and by several independent reviewers. In light of the reviews (below this email), we would like to invite the resubmission of a significantly-revised version that takes into account the reviewers' comments.

While we realize that the two reviewers differ substantially in their overall assessment of the manuscript, for the revision we would like you to pay great attention to fully address the two major points listed by reviewer 1:

1) Analysis of lab hybrids

2) Analysis of VSG content and comparison to phylogeny

We cannot make any decision about publication until we have seen the revised manuscript and your response to the reviewers' comments. Your revised manuscript is also likely to be sent to reviewers for further evaluation.

Sincerely,

Tim Nicolai Siegel, Ph.D

Guest Editor

PLOS Pathogens

David Horn

Section Editor

PLOS Pathogens

Kasturi Haldar

Editor-in-Chief

PLOS Pathogens

orcid.org/0000-0001-5065-158X

Michael Malim

Editor-in-Chief

PLOS Pathogens

orcid.org/0000-0002-7699-2064

Reviewer's Responses to Questions

**Part I - Summary**

Reviewer #1: Kay et al. address the question of genetic exchange in Trypanosoma brucei (and closely related species) using data from whole genome sequencing and assembly. Some of the sequencing data are new to this study, most are from previously published work, but all of the analyses are novel. Using data from lab hybrids, the authors present evidence of chromosomal cross-overs and post-meiotic ploidy change that have important implications for the largely cryptic mechanisms of genetic exchange in this species. They confirm previously proposed patterns of kDNA inheritance. These are then used in an analysis of VSG gene and kDNA maxi/minicircle composition in lab hybrids and field isolates, concluding that there is evidence of genetic exchange in these data even between relatively distantly related strains.

The mechanism of genetic exchange in T. brucei is an important and active topic and I found the addition of support for an emerging model of meiosis very interesting. The findings on kDNA and VSG inheritance are largely confirmatory, but I think constitute the best evidence to date. Similarly, there is existing support for genetic exchange (at least in T. brucei brucei and T. brucei rhodesiense), but additional support would be an important finding. However, sequencing depth, quality and assembly create a substantial source of potential artefact to the analysis and I don’t think this is currently adequately addressed. This particularly affects the analysis of VSGs, although it also has an expected influence on k-mer and SNP analysis. As such, in my opinion the data presented don’t currently support the major conclusions of the manuscript.

Reviewer #2: The manuscript by Kay et al presents an in-depth genomic analysis of several T. brucei isolates, including those from a genetic cross (parentals + descendents + siblings), which gave the authors important information on how recombination and gamete fusion occurs in the fly, as observed by the genomic evidence, and how this is correlated with the observed outcomes (diploids, triploids).

The manuscript is well written, complete and providing very important data for the community of scientists working in kinetoplastids. Important information is provided on how kDNA is passed onto progeny after a cross or through hybridization. Also very important information is provided on how VSGs, and MVSGs are shuffled in a cross.

Also very important information arises from use of all these observations on genetic hallmarks of recombination, and inheritance to field isolates from different outbreaks, and along large time spans (years).

I'd say this is a landmark paper that will guide the field for a number of years to come.

I congratulate the authors as I've have a very hard time finding _any_ suggestion to make, or a typo or mistake to highlight.

**Part II – Major Issues: Key Experiments Required for Acceptance**

Reviewer #1: 1) Analysis of lab hybrids

I don’t think the distribution of K-mers in F1R1 supports the authors’ interpretation of selfing. At least, I can find no proportion of Pop A and Pop B that would produce the distribution shown. If selfing is the cause, then the k-mer peak from additional genes in Pop B and the k-mer peak from genes not present in Pop B should be symmetrical around the 1x peak (with the shift in both directions dependent on the proportion of B). Isn’t it more likely that this distribution represents a mix of 2 clones or the presence of contaminating DNA in the sequencing? If the authors want to conclude that this is selfing, I think they would need to show that a ratio of A/B exists that can produce the distribution observed (I was unable to find one) and also rule out other easier explanations. For all of the data in Fig 1, it would be really useful for the inferred integer copy numbers be put on the distributions, so the match/mismatch to expectation can be assessed.

I also didn’t understand the SNP frequency histograms. Heterozygous alleles give SNP frequency modes at 0.5 (as per parents and F1G2 and F1Y4N). Alleles with greater ploidy should show peaks at rational frequencies (e.g. 1/3 and 2/3 for triploids, 1/4, 2/4, 3/4 for tetraploids, etc.). Again, the existence of irrational SNP frequency modes in F1R1 seems to me to be evidence against the author’s conclusion that this is selfing (which would still give rational modes). The authors attribute the lowest SNP frequency modes to the lowest k-mer modes, which doesn’t seem to make sense. How can a monoallelic gene have a SNP frequency? Why would 2x genes contribute to a peak in SNPs not at 0.5? Wouldn’t it make more sense for both peaks at 1/3 and 2/3 in F1R3N to come from the 3x genes? There is something about these data that I am not understanding as presented.

“J10 apparently having greater heterozygosity than 1738, as it has a more prominent haploid peak in the kmer plot, and more heterozygous sites in the SNP plot”. Firstly, the haploid peak for 1738 is obscured by the overlapping diploid peak, so the size cannot be assessed without fitting. More importantly, if the read depth or sequencing length/quality were considerably lower for 1738 than J10 (as one might infer from the peak spreads in Fig 1) this could greatly affect both the chances of assembly of haploid areas of the genome and the calling of SNPs (versus sequencing errors). This will have knock-on affects for both VSG set analysis and introgression analysis. Note that this won’t have the same affect on all parts of the genome, so unfortunately I don’t think metrics such as N50 or total assembled sequence will adequately account for possible artefact, here.

There is a big jump in analysis to produce the schematics in Fig 2, but the data used to arrive at these conclusions are not presented and no indication of confidence is provided. These are important inferences for the work, so some attempt needs to be made to show readers how these conclusions have been reached. How have calls been made about when a block is from one of other parent or mixed? How many SNPs of what confidence support each crossover event? There are established tests for detecting introgression (e.g. the D- and S*-statistics), but these don’t appear to have been applied here. At the least, it would be really useful to see a map of the density and classification of SNPs across the chromosomes in Supplement and a fuller explanation of how this has been parsed to produce Fig 2.

2) Analysis of VSG content and comparison to phylogeny

The authors need to be clearer that VSG numbers reflect assembly and detection not just representation in the genomes. Unless these lines have an order of magnitude fewer VSG genes than the genomes of lines mostly studied in the lab, then all of these sets are only a very small proportion of the full content. “Each clone had inherited approximately half its VSGs from each parent, with a bias to inheritance from J10 (paired t test, p = 0.0335, 95% CI), perhaps because J10 had more VSGs than 1738.” [line 258] suffers from the same artefact as the k-mer analysis, and it needs to be clear that “perhaps because we were able to assemble fewer VSG from this 1738” is likely more accurate. Similar for “Inheritance of these arrays in individual hybrid clones follows the same pattern” [line 277].

For this reason, in my opinion it is vital that shared VSG content (in analysis of hybrids and field isolates) be expressed as some measure of set-similarity, not as absolute numbers. For example, sharing only 12 VSG could either be a very low amount of overlap, or mean that every single VSG of 12 detected in one strain was also found in another strain. This is a particular issue when it comes to the analysis of “distantly related lineages” (Figs. 5 and 6). The analysis of linkages is expected to be substantially influenced by the completeness of the VSG/minicircle assembly. Indeed, all of the links highlighted in the text are between strains with some of the greatest number of assembled VSG/minicircles and I think a more convincing analysis is needed (ideally with some statistical support) to demonstrate this is significantly more than can be explained by assembly completeness.

The finding of discordance between kDNA maxicircle phylogeny and VSG/minicircle content for 2 isolates H865, H879 is really interesting, and suggestive. However, the Mx1 clade has a posterior probability of only 0.34 representing really weak support in this area of the tree and there is no confidence at all given for the clustering of H865 and H879 into type ‘c’ VSG/minicircle content. Note that both of these classifications need to be well-supported for the hypothesis of hybrid origin to have support, so proposing that this represents “compelling evidence” is a considerable over-reach. Either additional support needs to be provided, or the conclusions with respect to these data need to be substantially revised.

Reviewer #2: The only major issue in this manuscript is the lack of accession numbers for all the sequenced genomes.

The "Availability of data and material" reads "All data generated or analysed during this study are included in this published article and its supplementary information files." However the raw FASTQ files were not part of the supplementary materials and I failed to see accession numbers for these data mentioned in the Table S1, or S1 File. This must be corrected before publishing.

**Part III – Minor Issues: Editorial and Data Presentation Modifications**

Reviewer #1: In the reference genomes, there are quite a few VSG in the chromosome cores. How many of the genes in the analysis of hybrids are from core regions?

How can the parental lines contain “Additional VSGs not found in either parent” as represented by white circles in Fig 3?

What does “95% CI” indicate in the reporting of statistical tests? A confidence interval would usually be placed on an estimate of a value. Is the meaning here that the authors have chosen α<0.05 as the threshold for significance? I would suggest that reporting of the p-value is sufficient.

“The longest contig in each cluster terminated downstream of the promoter in telomeric repeats (TTAGGG) and/or a VSG (with the exception of cluster 6)” [line 290]. Is really unclear as written. Are contigs with just a promoter and telomeric repeat being considered MES here? This doesn’t seem right.

I found having the strains presented in a different order in Fig.5a and 5b unhelpful and counter to the point of the figure, which is to compare phylogeny with VSG/minicircle content. Could the data be reordered or a tanglegram be included to aid navigation?

I didn’t find that Fig 6 added very much to understanding, as it mostly represents a subset of the analysis presented in Fig 5. I would recommend it is moved to the Supplement.

Reviewer #2: I have not identified _any_ minor issue in this manuscript. Congrats to the authors!

Perhaps my only suggestion would be to include a graphic or schematic to summarize the observations on how nuclear, maxicircles, minicircles, VSGs, MVSGs are inherited after a cross, hybridization, etc. And which scenarios would produce different ploidies. I think this will help wrap up the manuscript coming full circle and presenting the readership an overview or guide to use genomic data to interpret these different scenarios (e.g. when sequencing a new isolate and trying to guess the clonal descent from a reference, or identifying putative parentals in extant sequenced genomes). This could be used also as a graphical abstract maybe.

PLOS authors have the option to publish the peer review history of their article (what does this mean?). If published, this will include your full peer review and any attached files.

Reviewer #1: No

Reviewer #2: No
---

## [Decision Letter · Decision Letter 1]

7 Dec 2021

Dear Prof. Gibson,

Thank you very much for submitting your manuscript "Signatures of hybridization in Trypanosoma brucei" for consideration at PLOS Pathogens. As with all papers reviewed by the journal, your manuscript was reviewed by members of the editorial board and by several independent reviewers. The reviewers appreciated the attention to an important topic. Based on the reviews, we are likely to accept this manuscript for publication, providing that you modify the manuscript according to the review recommendations.

Sincerely,

Tim Nicolai Siegel, Ph.D

Associate Editor

PLOS Pathogens

David Horn

Section Editor

PLOS Pathogens

Kasturi Haldar

Editor-in-Chief

PLOS Pathogens

orcid.org/0000-0001-5065-158X

Michael Malim

Editor-in-Chief

PLOS Pathogens

orcid.org/0000-0002-7699-2064

Reviewer Comments (if any, and for reference):

Reviewer's Responses to Questions

**Part I - Summary**

Reviewer #1: Thanks to the authors for engaging with my comments, particularly in the light of divergent opinion between the 2 reviewers.

F1R1 k-mer and SNP results

It was good to see that on re-analysis the authors agree that the previous hypothesis (mixture of 2 diploids with selfing) didn’t fit the data. I found the new analysis really helpful in seeing how the authors had parsed the data and the sequencing mixing and smudgeplot analyses are good additions. I’m still not convinced that a mix of diploid and triploid populations represents the only viable interpretation of these data (wouldn’t a single population in which only a single/few chromosomes are triploid also fit?), but the revised text doesn’t claim this as the only possibility and the data certainly seem compatible with such a situation.

SNP frequency histograms

I don’t want to be difficult here, and I apologize in advance if this is just a lack of understanding on my part, but the interpretation of the SNP histograms in Fig.1 still doesn’t make sense to me. The authors begin their rebuttal of my concern “Our SNP analysis considered only the mapped chromosomal cores [...] so there should not be any monoallelic genes”. However, their interpretation of the k-mer peaks in Fig.1 explicitly requires the existence of monoallelic genes which are marked red in all of the panels (a-f) and marked ‘1x’ in a,b,c and e. The figure still seems to mis-assign the SNP frequency peaks to the gene copy numbers. For example, F1R3N (e) shows data from a triploid strain expected to contain 1x copy genes (e.g. VSG), 2x (fewer as per lower k-mer peak) and 3x genes (blue dot in fig) as per the authors’ schematic. Haploid (1x) genes don’t have homologs in the same strain, so any read differences are just down to sequencing errors, which should to my understanding give ‘SNP frequency’ peaks near 0 or 1 (i.e. outside of the display range). Diploid genes can either be homozygous for a SNP (again 0 or 1) or heterozygous resulting in peak at 0.5. This peak appears to be too low to be detected in Fig. 1e. Triploid genes could again be homozygous at a SNP position (0 or 1) or 2:1 heterozygous (peaks at 1/3 and 2/3) or 1:1:1 heterozygous (peak at 1/3). The authors instead attribute the peak at 1/3 to the 1x genes and that at 2/3 to 2x. This still seems incorrect. It is possible I am badly misinterpreting these data, but I’m afraid the rebuttal didn’t clear this up for me.

Analysis of VSG content

Please note that my concern here was from the influence of completeness of VSG /assembly/ not read depth. While read depth could be a contributor to incompleteness it is not the only factor, so while the comparison of number of reads and number of VSGs was welcome it doesn’t really address my concern here. I am still of the opinion that analysis of the set-similarity rather than absolute numbers of VSG would be a much more suitable method here and offset many concerns about the influence of assembly completeness. However, I note that Reviewer 2 did not match any of my concerns and the authors have helpfully modified some of the text around these data, so I feel it is the authors’ decision whether or not to act here.

Minor points

All minor points raised have been addressed.

**Part II – Major Issues: Key Experiments Required for Acceptance**

Reviewer #1: I am of the opinion that the SNP frequency histograms (see above) still require correction or explanation before acceptance.

**Part III – Minor Issues: Editorial and Data Presentation Modifications**

Reviewer #1: All minor points raised have been addressed.

PLOS authors have the option to publish the peer review history of their article (what does this mean?). If published, this will include your full peer review and any attached files.

Reviewer #1: No

Figure Files:

Data Requirements:

Reproducibility:

References:

---

## [Decision Letter · Decision Letter 2]

22 Jan 2022

Dear Prof. Gibson,

We are pleased to inform you that your manuscript 'Signatures of hybridization in Trypanosoma brucei' has been provisionally accepted for publication in PLOS Pathogens.

Best regards,

Tim Nicolai Siegel, Ph.D

Associate Editor

PLOS Pathogens

David Horn

Section Editor

PLOS Pathogens

Kasturi Haldar

Editor-in-Chief

PLOS Pathogens

orcid.org/0000-0001-5065-158X

Michael Malim

Editor-in-Chief

PLOS Pathogens

orcid.org/0000-0002-7699-2064

Reviewer Comments (if any, and for reference):

Reviewer's Responses to Questions

**Part I - Summary**

Reviewer #1: Thanks to authors for taking another look at this. Modifications to the SNP peak assignments and also the cartoon mean that these now match.

**Part II – Major Issues: Key Experiments Required for Acceptance**

Reviewer #1: (No Response)

**Part III – Minor Issues: Editorial and Data Presentation Modifications**

Reviewer #1: (No Response)

PLOS authors have the option to publish the peer review history of their article (what does this mean?). If published, this will include your full peer review and any attached files.

Reviewer #1: No

---

## [Editor Report · Acceptance letter]

4 Feb 2022

Dear Prof. Gibson,

We are delighted to inform you that your manuscript, "Signatures of hybridization in Trypanosoma brucei," has been formally accepted for publication in PLOS Pathogens.

Best regards,

Kasturi Haldar

Editor-in-Chief

PLOS Pathogens

orcid.org/0000-0001-5065-158X

Michael Malim

Editor-in-Chief

PLOS Pathogens

orcid.org/0000-0002-7699-2064